# Feasibility of Mechanical Pollination in Tree Fruit and Nut Crops: A Review

Alieta Eyles [1],*, Dugald C. Close [1], Steve R. Quarrell [1], Geoff R. Allen [1], Cameron J. Spurr [2], Kara M. Barry [1], Matthew D. Whiting [3] and Alistair J. Gracie [1]

1   Tasmanian Institute of Agriculture, University of Tasmania, Hobart 7001, Australia; dugald.close@utas.edu.au (D.C.C.); stephen.quarrell@utas.edu.au (S.R.Q.); geoff.allen@utas.edu.au (G.R.A.); kara.barry@utas.edu.au (K.M.B.); alistair.gracie@utas.edu.au (A.J.G.)
2   Seed Purity, Derwent Avenue, Margate 7054, Australia; cspurr@seedpurity.com
3   Department of Horticulture, Washington State University, Pullman, WA 99164, USA; mdwhiting@wsu.edu
*   Correspondence: aeyles@utas.edu.au; Tel.: +61-326-622-668

**Abstract:** Pollination is essential for the production of most fruit and nut crops, yet it is often a limiting factor for both yield and product quality. Mechanical pollination (MP) systems offer the potential to increase productivity of a broad range of horticultural fruit and nut crops, and to manage the risk of reliance on current insect pollination services. To date, commercial MP systems have been developed for only a few crops (e.g., kiwifruit and date palm), suggesting that innovation in the use of MP systems has been stymied. Here, we review published and 'grey' literature to investigate the feasibility of MP systems of economically important tree fruit and nut crops. This review found that, whilst MP systems are a commercial reality for a wider range of fruit crops (e.g., sweet cherry) than nut crops (e.g., almond), promising results have been achieved at the experimental scale. Further we identified that the key barriers for progressing MP systems more widely include knowledge gaps in pollination biology, particularly of emerging fruit and nut species that are grown outside their native distributions, and access to proprietorial knowledge gained by commercial operators. What continues to remain unclear is detailed knowledge of the commercial development of MP systems and therefore, the opportunities to apply this knowledge to other tree crops where effective pollination limits yield and quality.

**Keywords:** pollination biology; pollen quality; pollinator; pollinisers; incompatibility

## 1. Introduction

Perennial crops are considered important contributors to food security despite only covering 4.2–4.7% of the total cropping area [1]. In 2020, the value of global production of fruit and nuts was USD 823 billion with the top five fruit crops being apples (87 Mt), oranges (78 Mt), grapes (77 Mt), mangoes (55 Mt) and apricots (41 Mt) [2]. Although considerably smaller in terms of global production, the tree nut industry is rapidly expanding. In 2019/2020, global production of tree nuts was estimated to be 4.2 Mt, of which almonds, walnuts, cashews, pistachios and hazelnuts contribute 31%, 21%, 17%, 14% and 12% of the market share, respectively [3].

Tree crops represent a long-term investment, bearing commercial yield several years after planting (e.g., at 3 years of age for sweet cherry and 10 years of age for pistachio [4]). Best-practice tree crop production utilises high-density monocultures of cultivars selected for high yields, fruit quality and disease resistance, managed within intensive high-input growing systems [5]. However, both yield and quality of these agricultural commodities can be severely limited by inadequate pollination [6,7]. Therefore, significant efforts in the science and management of pollination are urgently warranted.

Efficient pollination and, conversely, pollination limitations or deficits, depend on both intrinsic and extrinsic factors (Figure 1, Table 1). Intrinsic factors include the availability

and quality of compatible pollen, the duration of stigma receptivity, and ovule longevity (reviewed in Howlett et al., 2015 [8]). Extrinsic factors include weather conditions and their effect on synchrony of flowering, pollinator activity, pollen quality and agronomic management practices regarding plant nutrition, orchard design, and pests and diseases (see reviews by Toledo-Hernandez et al., 2017 [9]; Pardo and Borges 2020 [10]). According to Hatfield and Prueger (2015) [11], "pollination is one of the most sensitive phenological stages to temperature extremes across all species" and, as such, productivity is also greatly affected by temperature extremes. Indeed, climatic effects have been demonstrated to be key determinants of pollination success and yield in apple and almond [12], and peach [13]. In addition to reducing yield, poor pollination has been consistently shown to affect crop quality including fruit symmetry (e.g., apple, [14]), mineral content and storability of apples [15], and oil and vitamin E content of almond [16].

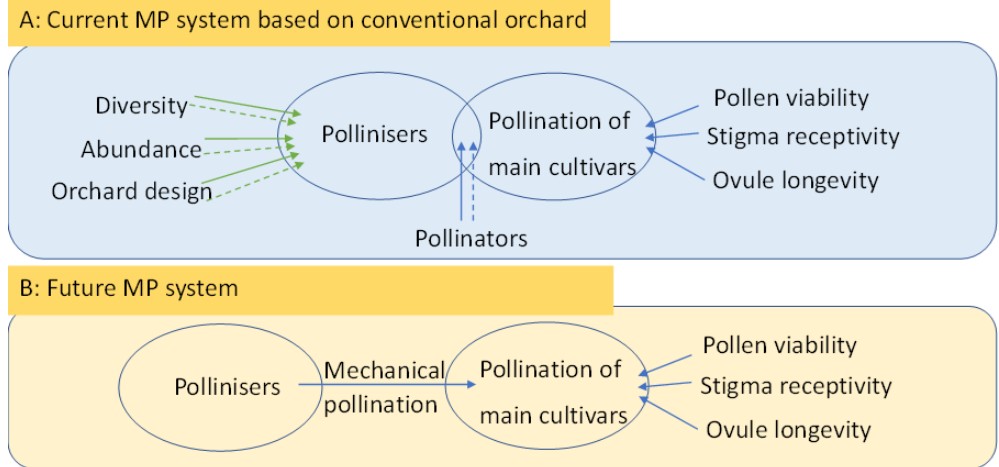

**Figure 1.** Conceptual diagram of the factors influencing cross-pollination of fruit and nut crops for a polliniser and pollinator model production system (**A**) with (solid line) or without (dashed line) a mechanical pollination (MP) system whereby factors with dotted lines would have reduced influence under a MP system, or (**B**) MP only where pollinisers are grown in a separate orchard. In both scenarios, the same environmental conditions and agronomic management practices would apply.

**Table 1.** Factors contributing to pollination deficiency in fruit and nut crops, and potential benefits of mechanical pollination (MP) systems.

| Factors Contributing to Pollination Deficiency | Benefits of Mechanical Pollination Systems |
| --- | --- |
| Adverse weather conditions may reduce insect activity, flowering time and flowering intensity. | Buffer against adverse weather conditions as MP can be applied day or night under sub-optimal weather conditions but not during rain events [17]. |
| Asynchrony of flowering—the overlap of pollen and stigma receptivity varies from year-to-year depending on environmental conditions [18]. For monoecious species—asynchrony of flowering of compatible cultivars. For dioecious species, asynchrony between staminate and pistillate blossoms. | Reduced reliance on pollinisers as pollination can occur when the main crop is ready. |

**Table 1.** *Cont.*

| Factors Contributing to Pollination Deficiency | Benefits of Mechanical Pollination Systems |
| --- | --- |
| Preference of honey bees for alternative nectar and/or pollen sources to that of the main crop. However, reduction in non-target forage sources for pollinators may reduce pollinator health to due reduced nutrition [19]. | Diversity of non-target forage sources can be promoted to support managed and native pollinators. |
| Crop species not being suited to honey bee pollination as flowers are not attractive to honey bees e.g., lack of nectar or poor quality nectar (e.g., pear [20]; kiwifruit [21]). | Reduce reliance on managed and native pollinators. |
| Pollinisers can produce a high percentage of sterile [22] or non-viable pollen [18]. Adverse weather conditions and agronomic practices (e.g., pesticides) can also reduce pollen viability [23]. | Pollen quality of stored pollen can be monitored and sourced from orchards producing fertile and viable pollen in a given season. |
| For established orchards, weaknesses in the pollination design, e.g., density (10–20%) and distribution (10–15 m range of compatible polliniser and ratio of male to female trees and in the appropriate wind direction for anemophilous species [24]. Little flexibility to test new pollinisers. | For established orchards, opportunity to improve pollination by applying new improved compatible pollinisers that could improve fruit size and quality (e.g., via xenia and metaxenia effects, [25]. For new orchards, reduction or elimination of polliniser that often has less commercial value of fruit than the main cultivar and can represent a risk if more prone to pest or disease pressure. |

　　　With the exception of parthenocarpic crops such as citrus, banana and common fig, and some self-fertile cultivars, most tree crops are cross-pollinated and are highly dependent on insects for successful pollination ([26]. Horticultural producers rely on both natural and managed insect pollination services to achieve seed and fruit set [27]. This has led to a heavy reliance on the European honey bee (*Apis mellifera*) from either wild colonies and/or managed hives to pollinate and improve productivity of commercial horticultural tree crops [28]. This dependence on a single pollinator species poses a major risk to fruit and nut production. Managed and wild bee populations are declining due to a suite of interacting factors including pesticides, pests, diseases, migratory beekeeping and landscape fragmentation [29]. Arguably, the most devastating causal factor has been the invasive parasitic mite *Varroa destructor*, which feeds directly on developing pupae and adult bees, and vectors bee viruses. Despite increased understanding of *Varroa* (reviewed in Traynor et al., 2020 [29]), identifying sustainable control options appears to be decades away (reviewed in Guichard et al., 2020 [30]).

　　　In the face of these challenges, there is an urgent need to find alternative approaches to improve pollination. Strategies that are being considered to reduce the risk of reliance on managed honey bees include the use of alternative pollinators (reviewed in Rader et al., 2020 [31]), in addition to breeding crops that are parthenocarpic or more self-compatible and self-fertile and, therefore, less reliant on biotic pollination ([32]. Further, although labour intensive and costly, manual pollination is also routinely undertaken for some high value species. In the case of vanilla, this is undertaken due to a lack of specialised pollinators outside of Central and South America [33]. Alternatively, a more cost-effective approach can be mechanical pollination (MP), which offers the potential to offset the limitations of natural pollen transfer systems including delivering consistent pollination rates; reduced risks associated with unfavourable environmental conditions and climate variability; and reduced reliance on current insect pollination services (Figure 1).

In this review, MP specifically refers to any type of pollination that uses a mechanical device to apply previously collected pollen to a target crop. All other forms of non-mechanical pollination systems, often described as 'assisted', 'controlled' or 'supplementary' pollination, are referred to as artificial pollination for clarity. Pinillos and Cuevas (2008) [34] reviewed artificial pollination in tree crop production and provided a comprehensive summary of key concepts in artificial pollination. Therefore, this review focuses on findings since 2008, with a specific focus on MP systems, drawing from the knowledge gained in artificial pollination studies where required. Furthermore, we found that the understanding of pollination biology has increased significantly in the past 15 years for some tree crops, particularly of emerging tree crops such as avocado [35] and macadamia [36,37], and even for established trees crops such as olives [38,39].We acknowledge the critical importance of pollinators; however, this topic is beyond the scope of the review and has been reviewed elsewhere [26].

In this review, we analyse and evaluate key components of MP systems including pollen collection, handling, storage and delivery, in addition to identifying possible risks linked to MP. We also discuss the future prospects of developing MP systems for economically important fruit and nut crops. We suggest that it is possible to determine the feasibility of MP systems in tree crops for which detailed knowledge of their pollination biology exists.

## 2. Benefits of Mechanical Pollination Systems

Table 2 represents a current list of studies that have investigated MP systems since 2008; however, potential exists for this list to only represent trials that were successful, given the challenges of publishing negative results. Nevertheless, MP was found to significantly increase fruit and nut set for eight different tree crops, with the exception of pistachio (Table 2). The majority of these studies applied pollen using handheld sprayers, suggesting that MP systems are still in their infancy. However, it is clear from the 'grey' sources (e.g., [17,40]) that there have been significant technological and engineering advances in the implementation of MP systems and that MP is a commercial reality for some crops, including kiwifruit (reviewed in Mu et al., 2018 [41]), sweet cherry [17], date palm and almond [42]). For example, specialised MP equipment is available for commercial sale for date palm (e.g., AgroPalm Machinery [43]) and, in New Zealand, multiple companies supply harvested pollen for the kiwifruit industry (e.g., Kiwifruit Vine Health [44]) or sell machinery to process, extract, dry, store and deliver kiwifruit pollen (e.g., Fraser Gear [45]).

**Table 2.** Examples of mechanical pollination (MP) in fruit and nut crops published since 2008 (only includes studies that used processed pollen or mechanical equipment). * 'grey' literature. # No information on when fruit set was assessed.

| Crop | Type of Application | Pollen Information | Pollen Carrier | Impact | References |
|---|---|---|---|---|---|
| Cacao (*Theobroma cacao*) | Blower | No pollen applied | None | Increased fruit yield by ~8% compared to natural pollination | [46] |
| Date palm (*Phoenix dactylifera*) | Handheld sprayer | Air dried. No other details given. Concentrations (0.5, 1.0, 1.5, 2 g $L^{-1}$) | Liquid carrier (water) (no benefit for 10% sucrose +1% agar) | Increased fruit set by 7% and 18% (assessed 7–8 weeks after pollination), depending on pollen concentration | [47] |
| Date palm (*Phoenix dactylifera*) | Handheld duster (attached to a 10 m boom) | Air dried No other details given | Dry carrier—wheat flour (1:5, 1:10, 1:15) | Manual pollination had higher fruit set (40–50%) than MP (30–40%), however yield was comparable. Fruit set was assessed 5 weeks after pollination | [48] |

**Table 2.** *Cont.*

| Crop | Type of Application | Pollen Information | Pollen Carrier | Impact | References |
|---|---|---|---|---|---|
| Date palm (*Phoenix dactylifera*) | Handheld sprayer (applied once) | Air dried and stored at 4 °C. | Liquid carrier—water (3 g L$^{-1}$) | Mature fruit set was higher in MP (86%) than manual pollination (69%) | [49] |
| Date palm (*Phoenix dactylifera*) | Handheld ducted fan blower (attached to a ~7 m boom) | Air dried then stored at 4 °C for 6 days Concentrations (0.5, 2.5, 5.5 g) Applied up to 3 times | No carrier or wheat flour (1:10—10% pollen and 90% wheat flour) | Fruit setting efficiency (assessed 8 weeks in the Kimri stag) of pollen applied above 2.5 g was comparable to manual pollination | [50] |
| Kiwifruit (*Actinidia deliciosa*) | Handheld blowers, backpack sprayer, tractor-mounted sprayer and hand pollination | 600 to 1200 g ha$^{-1}$ | Dry (Lycopodium) and liquid (PollenAid) carrier | Timing of pollen application more important than pollen amount. Yield increased by up to 12% | [51] |
| Kiwifruit (*Actinidia deliciosa*) | Handheld sprayer | Pollen dipped in 99.5% acetone and stored at 4 °C *sans* acetone | Liquid pollen carrier (sodium chloride, Arabic gum, PGDO) (4 g L$^{-1}$) | No control Freshly made carriers increased seed number better than those stored for up to 5 h | [52] |
| Kiwifruit (*Actinidia deliciosa*) | Handheld pollen blower and pollen dispensers | Pollen blower: 400 to 1600 g ha$^{-1}$ Pollen dispenser: 4 g per hour for 4 h on 4 days | No carrier on the day of application (but applied on the day before application) | No relationship observed between pollen concentration and seed number | [53] * |
| Kiwifruit (*Actinidia deliciosa*) | Handheld blowers, tractor-mounted sprayer (9 treatments) | 600 g ha$^{-1}$ | Liquid and dry carriers (Lycopdium (45%:55%; pollen; Lycopdium), PollenAid (12 g L$^{-1}$) | Lycopodium may have a drying effect | [54] |
| Kiwifruit (*Actinidia deliciosa*) | Pollen dispenser (Flying Doctor®) using *Bombus terrestris* (8–9 hives ha$^{-1}$) | Pollen dispenser (54–60 g ha$^{-1}$ day$^{-1}$) Manual pollination (250 g ha$^{-1}$) | No carrier | No detail of fruit set but seeds per fruit and individual fruit weight were comparable | [55] |
| Kiwifruit (*Actinidia deliciosa*) | Sprayer (one spray application —no details given) | Pollen vacuumed from male flowers and stored at −20 °C | Liquid carrier (water) (3 g L$^{-1}$) | Bee pollination (0.93) resulted in higher yield than artificial pollination (0.65) | [21] |
| Kiwifruit (*Actinidia deliciosa*) | Air-liquid nozzle spraying vs. electric sprayer | | Liquid carrier | Air nozzle spraying (87% fruit set [#]); Electric sprayer (74%) | [56] |
| Japanese pear (*Pyrus pyrifolia*) | Direct hand application and electromotive-style sprayer | Refined with acetone and stored at −30 °C. Germination 35–45% | Liquid carriers (containing agar, xanthan gum, pectin methylesterase or polygalacturonase) | Variable responses however, initial fruit set (60–86%) with liquid carrier was comparable or lower than hand pollination (74–94%) | [57] |
| Olive (*Olea europaea*) | Mechanical blower applied twice | Stored at 4 °C for a few days. Germination 35 to 68% 2 g per plant | No carrier | Fruit set [#] of MP (15%) was higher than control (3.7%) | [58] |
| Olive (*Olea europaea*) | Powder duster (up to four applications) | Stored for 1 year at −20 °C 80 g ha$^{-1}$ | No carrier | Improved yields were only observed in fruit bearing years and none in unproductive years | [59] |
| Sweet cherry (*Prunus avium*) | Electrostatic and airblast sprayer Two applications | 36, 72, 144 g ha$^{-1}$ | Wet carrier (patented recipe) | 10 trials: highly variable results depending on cultivar and year ranging from 0% up to 20% increase in fruit set [#]. Commercial orchards had 4 to 5 hives per acre. | [17] * |

**Table 2.** *Cont.*

| Crop | Type of Application | Pollen Information | Pollen Carrier | Impact | References |
|---|---|---|---|---|---|
| Almond (*Prunus dulcis*) | Hand-held and backpack-mounted sprayers (two applications) | Bee-collected pollen stored at −20 °C. No other details given. | Liquid carrier (recipe in Hopping and Simpson, 1982) | No details given except that MP was not equivalent to hand pollination. | [60] |
| Almond (*Prunus dulcis*) | Electrostatic sprayer (two applications) | Germination > 85% No other details given. Concentrations (0, 59 and 175 g ha$^{-1}$) | Liquid carrier (Pollen-tech®) | Year 2014: Applied twice (30–40% and 50–60% bloom)—nut set not improved by treatment Year 2015: Applied once at 60% bloom—MP achieved 17% nut set compared to 57% for combined bee and MP | [61] * |
| Hazelnut (*Corylus avellana*) | Handheld blower | Stored −12 °C for 2 weeks Germination (46–52%) Concentration—details not clear | Dry carrier−1% Lycopodium spore | increased yield (37%) than wind control | [62] |
| Hazelnut (*Corylus avellana*) | Backpack sprayer or direct hand application | Stored at −20 °C 30 g ha$^{-1}$ Germination ranged from 7.5 to 34.5% | Liquid carrier (PollenAid or 10% sucrose solution + 0.5% xanthan gum, 0.02% boric acid) | Increased fruit set (50%) compared to hand application | [63] |
| Hazelnut (*Corylus avellana*) | Mist blower | Stored at −20 °C 150 g ha$^{-1}$ | Liquid carrier (10% sucrose solution + 1% agar, 0.02% boric acid | Did not assess fruit set but stigma receptivity and ovule health | [64] |
| Pistachio (*Pistacia vera*) | Handheld sprayer | No details given on storage. | Liquid carrier | Fruit set [#] was generally lower when applied by MP than control | [65] |
| Pistachio (*Pistacia vera*) | Handheld sprayer (3 applications) | No details on storage conditions. | Liquid carrier (agar, zinc sulphate) | Variable responses but the addition of pollen did not necessarily improve mature fruit set | [66] |
| Pistachio (*Pistacia vera*) | Handheld sprayer (3 applications) | No details on storage conditions. | Liquid carrier (agar, boric acid) | MP decreased mature fruit set (3–7%) compared to open pollination (7–10%) | [67] |

Innovative pollination strategies such as MP systems have the potential in their most basic form to supplement existing pollination services, whereas more sophisticated MP systems can offer additional benefits (Figure 1). The level of complexity of the MP system needed would depend on the production system of the tree crop. For example, implementation of a MP system would not necessarily mean the exclusion of managed pollinators. Instead, hybrid MP systems can be implemented in which traditional insect-derived pollination services are supplemented by direct pollination with MP systems or via insects re-distributing pollen applied via MP, as currently practised in kiwifruit [53]. In the long term, a fully realised MP system would be based on the total replacement of polliniser (pollen donor) trees within commercial production orchards with specialist orchards established for the sole purpose of pollen production (Figure 1). This is currently being realised in the USA, where orchards were planted in 2017/2018 for the sole production of sweet cherry pollen with pruning and training strategies designed specifically to promote high flower and pollen yield [17].

In addition to reducing the reliance on managed pollinators for pollination, MP systems offer flexibility when making strategic decisions about the genetic properties of the pollen applied (Table 1, Figure 1). In particular, the pollen distributed may have desirable xenic and meta-xenic effects on fruit and nut characteristics, including size, shape colour, flavour and nutrient quality, and the fruit-ripening period (reviewed in Yang et al., 2020 [25]). For example, an increase of up to 45% in nut weight was reported for one of

four pollinisers studied in hazelnuts due to xenia effects [68]. Similarly, cross-pollination increased the total soluble solid and phenolic content, contributing to higher antioxidant content of 'Qicheng52' navel oranges when pollinated with Murcott tangor pollen [69].

## 3. Processes of Mechanical Pollination

Mechanical pollination typically involves four key steps: (1) pollen collection, (2) pollen handling and drying, (3) pollen storage, and (4) pollen delivery (Figure 2). Pinillos and Cuevas (2008) [34] have already covered key concepts for each step in their review; therefore, the next section largely provides summaries and, where relevant, updated examples since 2008.

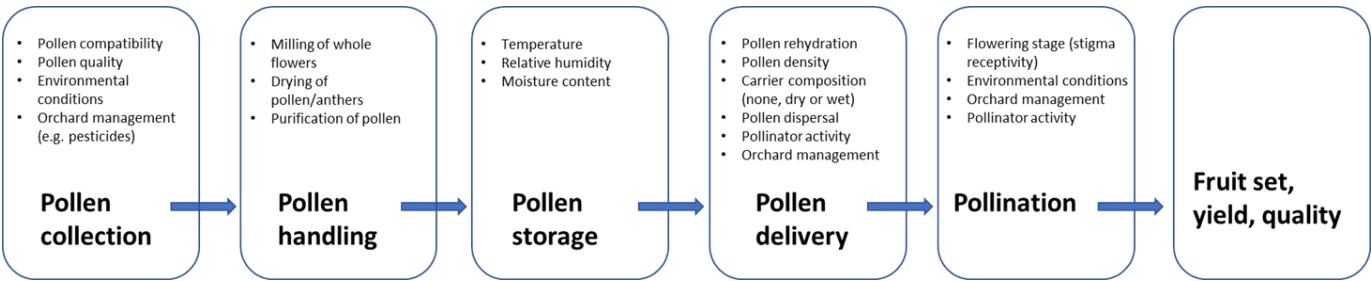

**Figure 2.** Key steps and factors in mechanical pollination systems (modified from Cacioppo et al., 2018 [58]).

### 3.1. Pollen Collection

Of paramount importance for an effective MP system is ensuring the efficient collection of high-quality pollen. Typically, pollen is harvested prior to anthesis to avoid contamination and to minimize loss via pollen shedding (Figure 2). Pollen quality can vary by genotype [17,18], weather conditions during flowering [70], flower maturity [71] and agronomic practices (e.g., pesticide application, [23,72]). A comparison of nine pear cultivars (*Pyrus communis*) grown in the same orchard found in vitro germination ranged from 3 to 71%, with values increasing to 12 to 88% in the subsequent year due to warmer weather [18]. Some commercial suppliers of kiwifruit pollen guarantee an in vitro germination rate of at least 80% (e.g., Pollen Plus [73]). Although such high germination rates are desirable for maximising the likelihood of success of MP systems, this may not be achievable and even not strictly necessary for some tree crops. Indeed, in hazelnut, a preliminary trial showed improved fruit set even using pollen with germination rates of 7.5 and 34.5% [63].

The idea of bees collecting pollen—by installing traps at the entrance of hives to collect pollen from the bee corbiculae—has persisted. However, recent studies have shown that the function of pollen collected by corbiculate bees (*Bombus impatiens* and *Apis mellifera*) was reduced compared to pollen collected by non-corbiculate bees (*Megachile rotundata* and *Halictus* spp. [74]). For corbiculate bees, packed pollen was shown to be 27% less effective in pollinating *Brassica rapa* ([74]), suggesting the limited usefulness of honey bee-collected pollen in MP systems.

Mechanized harvesting methods have been successfully developed for anemophilous species that produce large amounts of easily released pollen. For example, a backpack-mounted pollen vacuum was reported to directly collect more than 500 cc of Douglas fir (*Pseudotsuga menziesii*) pollen per hour under good shedding conditions [75]. Similarly, a modified vacuum method was tested to collect large quantities of pollen from cannabis (*Cannabis sativa*), another anemophilous species [76].

In contrast, mechanical harvesting of pollen from entomophilous species continues to be challenging, partly as these species produce less abundant amounts of 'sticky' pollen (e.g., almond [77]). Therefore, for these species, pollen collection continues to rely on expensive manual methods that potentially can pose a major restriction on the use of MP

systems. However, the number of commercial pollen suppliers that are available suggests that manual harvesting of pollen is sufficiently profitable, which is perhaps not surprising given that MP systems support high value tree crops. In 2016, a commercial pollen supplier reported sales of 2 t of fruit pollen per year [78]. In 2018, the cost of kiwifruit pollen in New Zealand was USD 3452 kg$^{-1}$, with a recommended application rate of 300–400 g ha$^{-1}$ at least twice during flowering (No. 1 Pollen [79]). Other companies listed have prices ranging between USD 1200 and 1300 kg$^{-1}$ for pure pollen from a variety of fruit trees and almond, with the average cost per ha ranging from USD 295 to 585 (e.g., Firman Pollen [80]).

### 3.2. Pollen Handling

In general, pollen handling involves the physical separation of anthers from flowers via milling and/or sieves, followed by the drying of anthers/pollen, with the final step typically involving the collection of purified pollen using a vacuum cyclone (e.g., Fraser Gear [45]). Specified levels of pollen purity can be obtained from pure to lower grades, depending on pollen application requirements for achieving the desired level of pollination and fruit/seed set. Some pollen suppliers offer a range of purity grades with lower grades of purity sold at lower prices (e.g., Firman Pollen [80]).

Since 2008, there have been major engineering advancements in pollen handling, with commercial companies currently selling specialised machinery to complete these activities for both anemophilous and entomophilous tree crop species. For example, a flower mill sold in New Zealand has the capacity to process 2 t of fresh kiwifruit flowers per day (e.g., Fraser Gear [45]), whereas an automated pollen extraction machine can produce 2.5 to 5 kg pollen from male flowers of date palm (e.g., AgroPalm Machinery [43]). Additionally, customised drying cabinets that allow for precise temperature and relative humidity control are also commercially available (e.g., Fraser Gear [45]).

### 3.3. Pollen Storage

MP systems rely on stored pollen. Commercial suppliers of pollen state that their pollen, when appropriately processed and managed, can remain viable for two to five years without significant loss (e.g., Pollen Pro [81]). Extensive knowledge on long-term storage of pollen already exists, driven by the critical relevance to breeding programs and genetic conservation. The majority of these studies focus on the effect of temperature on pollen viability for both short- and long-term storage (Table S1). In general, floral pollen stored at 4 °C for one to three months remains viable, depending on the fruit tree species. Lower storage temperatures such as −20 °C are required to increase pollen longevity over periods longer than one to three months (Table S1).

What has been less studied is the impact of drying in maintaining pollen viability during long-term storage [82]. Delays in drying can result in a reduction in pollen quality and complete loss of viability. In nature, before pollen is dispersed, it generally undergoes a controlled process of dehydration within the anther. Pollen in a partially dehydrated state has reduced metabolic activity and is longer-lived and better able to tolerate further desiccation during dispersal [83].

The ideal drying conditions (humidity, temperature and air flow rate, and time) for lowering the water content of pollen has been shown to vary between species. Some reports indicate a moisture optimum of 15%, whereas higher water concentrations (above 30%) may result in rapid deterioration [84]. Pollen cells with high moisture levels do not survive freezing storage, presumably due to the formation of lethal intracellular ice and subsequent membrane rupture [85]. However, there appears to be a minimum moisture content below which longevity is not improved and, in fact, may adversely affect pollen viability [84].

Pollen can be classified according to the number of cells within the grain at maturity or anthesis [34]. Seventy percent of flowering species have bicellular pollen, whereas the other 30% have tricellular pollen [86]. Tree crops, both fruit and nut [87], produce bicellular pollen, which is more resilient to drying than tricellular pollen. Bicellular pollen can generally withstand drying to moisture contents of less than 11.1% dry weight [88].

In practice, this means that this type of pollen is more easily stored than trinucleate pollen, which is more differentiated, more metabolically active and has short longevity relative to binucleate pollen [83]. However, there is little information on the exact moisture content of pollen that is required to maintain pollen longevity during storage (e.g., *Nothofagus alpina* [82]), possibly in part due to the challenges of accurately measuring moisture content for such small quantities. Rapid methods based on NIR technology have been used to quantify moisture content of rice grain and maize seeds [89,90]. We argue that accurate measurements of moisture content of pollen are similarly required to provide a quick indicator of potential pollen quality, particularly of large volumes, prior to long-term storage.

### 3.4. Pollen Delivery

A range of technologies are currently available or under development to deliver pollen (Table 3). In its simplest form, MP can be performed using 'pollen dispensers' containing pre-collected compatible pollen fitted into the opening of a hive of honey bees or bumble bees [53]. Bees are forced to walk through the pollen dispensers, become covered with pollen during their exit from the hive, and deliver this pollen when foraging on the target flowers. Improved design of these dispensers reduces pollen wastage [20,53,91] and allows the dissemination of biological agents (e.g., Prestop® Mix) to control fungal infections such as botrytis and sclerotinia (e.g., apples; [92]). Although pollen dispensers are inexpensive and useful, this option does not overcome the inherent risk of reliance on a single species pollinator.

**Table 3.** Summary of pollen delivery options (from the simple to the more complicated designs). * Once mixed in wet carrier, pollen viability is limited to 100 min (Whiting 2019).

| Delivery Method | Carrier | Comments | Reference |
|---|---|---|---|
| Manual (Hand pollination with brush, glass rod, feather etc.) | No | Time consuming and laborious | See references cited in Table S1 |
| Pollen-containing dispenser (which is located at the entrance of beehive.) | Optional (dry) | Reliance on activity of bee—if weather conditions poor than low pollinator activity. Allows targeted delivery. Commercial option. | e.g., Flying Doctors [20,55,91] |
| Hand-held sprayers/blowers | Optional (dry and wet *) | Bees can help redistribute pollen if using a dry carrier. Commercial option. | See references cited in Table 2 |
| Mobile sprayers/blowers (e.g., quadbikes) | Optional (dry and wet *) | Bees can help redistribute pollen if using a dry carrier. Commercial option. | See references cited in Table 2 |
| Pollen application sprayers with booms and pressure vessel | Optional (dry and wet *) | Does not work in rain. Viable commercial option. Most based on electrostatic spraying. | Examples:<br>• On Target Spray Systems® [17,94]<br>• Pollensmart ® [95]<br>• LectroBlast (Progressive Ag Inc. [96]) |
| Unmanned aerial vehicles (UAVs) (Drones) (aerial delivery of pollen) | Dry only if used. | Technology is new; however, this service is commercially available. Limited capacity on amount of pollen it can transport (e.g., 5 kg). Does not work in rain. Commercial option. | e.g., Dropcopter [97] No peer-reviewed paper found on pollination but see Zhang et al. (2019) [98] on UAVs in orchard management. |
| Robotics e.g., 1: platform mounted manifold spray nozzle; e.g., 2: robotic bees | Optional (dry and wet *) | Autonomous, precision operation. Technology is new. | e.g., Kiwifruit, [93,99,100] |

The majority of peer-reviewed studies have largely tested handheld sprayers/blowers to test the feasibility of MP systems, given the smaller scale of research studies (Table 2). However, the efficacy of MP systems will also need to be tested at the orchard scale level us-

ing boom sprayers mounted on vehicles, as reported in the 'grey' literature for almond [61], kiwifruit [56] and sweet cherry [17]. A mechatronic (computer controlled, electromechanical) prototype that utilises sensory information to make 'intelligent', calculated decisions in real time, such as flower detection and autonomous operation, was tested on kiwifruit with promising results—the robotic system had a spraying accuracy of 80% of flowers [93] (Duke et al., 2017).

A major concern with most pollen delivery methods is the lack of precision and utilisation. According to Goodwin and McBrydie (2013) [53], with general broadcast methods, most of the pollen (>99%) never reaches the tiny stigmatic surface (e.g., ~ 1 mm$^2$ for sweet cherry [101]). Strategies have been developed to improve pollen utilisation, reduce pollen dosage and/or improve precision in delivery. For example, the efficacy of air delivery methods used in kiwifruit was improved by the presence of honey bees that redistribute the applied pollen onto pistillate flowers [53]. Alternatively, the effectiveness of wet spray pollination methods has been shown to be increased by the use of electrostatic forces, whereby charged pollen grains settle more effectively on the stigma than uncharged pollen [102].

For some tree crops in which only a fraction of the flowers is receptive at the time of application, multiple applications have been shown to be required to improve fruit set (Table 2). In these cases, the benefits gained from MP may be offset by the relatively high cost of pollen. A recent study comparing the pollination requirement of two kiwifruit varieties found that 'Zesy002' required around 66% fewer pollen grains to set fruit than 'Hayward' [103]. These findings suggest that it is possible to tailor MP systems to both save cost and increase the efficiency of pollen application. Further, field trials using a boom pollen delivery method found that single applications could improve fruit set in sweet cherry. However, the timing was critical and needed to coincide with a large population of recently opened flowers to achieve a commercially acceptable crop [17]. More importantly, these trials concluded that MP systems are still constrained by environmental conditions, as no improvements in fruit set were observed when pollen was applied during rain [17].

## 4. Potential Risks of Mechanical Pollination

### 4.1. Pollen Transmission of Bacteria and Viruses

The intrinsic feature of MP being the collection and transfer of pollen means there is a need to consider the potential role of pollen as a vector of diseases. Viruses and viroids are the largest concern for pollen-vectored pathogens. Of the approximately 100 bacterial species known to be plant pathogens, only three species have been suggested to involve pollen-mediated transmission in perennial crops i.e., *Xanthomonas arboricola* pv. *juglandis*, *Erwinia amylovora* and *Pseudomonas syringae* [104]. However, until recently, evidence of the importance of pollen and pollination for the dissemination of these pathogens was largely based on experimental conditions via artificial inoculations rather than direct detection of the pathogen in field tests [105]. Using a mutant with a green fluorescent protein, Donati et al., (2018) [104] was able to show that *P. syringae* pv. *actinidiae* (PSA) was able to directly colonize kiwifruit anthers epiphytically and endophytically, resulting in the production of contaminated pollen that could transmit this pathogen to healthy plants. It is possible, but not confirmed, that the incursion of PSA into New Zealand, which led to severe damage to the kiwifruit industry, may have been pollen-borne [106].

Of over 1000 plant viruses recognised by the International Committee on Taxonomy of Viruses, at least 46 plant viruses have been reported as being pollen-transmitted [107]. Several of these have been reported in perennial fruit crop hosts, including *Rubus* spp., *Prunus* spp. and *Vaccinium* spp. [108]. Transmission of viruses can occur through the pollinated flower (horizontal transmission, e.g., Raspberry bushy dwarf virus [107] or through the developed seed (vertical transmission, e.g., Prune dwarf virus [109]).

The biosecurity risk posed by the commercial movement of pollen between orchards, regions and countries means that quality assurance protocols will need to be developed to ensure traceability, purity and freedom from pathogenic contamination. The detection and

quantification of viruses using robust assays that are species specific and highly sensitive (e.g., RT-qPCR assay [109]) will be required to aid in the implementation of biosecurity protocols. From the grey literature, compliance appears to be voluntary. For example, suppliers of pollen of sweet cherry sold for mechanical pollination in the United States confirm that samples are tested for viral pathogens, e.g., cherry leafroll virus and bacteria (e.g., PollenPro [81]), and, in New Zealand, the risk of *P. syringae* in the kiwifruit industry is managed by protocols implemented by Kiwifruit Vine Health (Kiwifruit Vine Health [44]), a dedicated biosecurity agent. Alternatively, individual orchardists could minimise their risk of disease dissemination by supplying their own pollen—a service offered by some pollen suppliers (e.g., No. 1 Pollen [79]).

*4.2. Reducing Yield and/or Quality*

Of the limited number of papers published on MP systems (Table 2), the majority of studies have reported improved crop yield. For some tree crops, achieving a higher fruit/nut set may not be desirable as resource limitations may negatively influence product size and quality [110]. For example, increased pollination success has been demonstrated to reduce macadamia nut size but not quality [111].

Arguably, high-value trees crops should face fewer resource limitations due to fertiliser inputs; nonetheless, high fruit set may still have negative impacts for some tree crops. For example, for tree fruit crops where fruit size is important, the development of a MP system would need to avoid inadvertently increasing thinning costs, particularly in varieties which are considered to be heavy cropping (e.g., apple [112]). Chemical and mechanical thinning is routinely undertaken to manage crop load to ensure fruit quality and marketable size, to reduce the risk of branches breaking due to heavy fruit loads and to prevent biennial bearing [112]. For the latter, strategic application of MP during the 'off' year may assist remedial management of trees that are in a cycle of biennial bearing by minimising any pollination deficit during off years, thereby helping break the biennial bearing cycle.

**5. What Factors to Consider When Predicting the Likelihood of Success of MP for Fruit and Nut Crops**

A successful MP system must be underpinned by a strong understanding of the pollination biology of the target species. This includes detailed knowledge of male and female flowering phenology of the cultivars as it relates to pollen collection, pollen handling and storage, and effective pollen delivery (how many times and at what stage(s) of development should pollinations be made). Although the majority of tree crops are cross-pollinated, even tree crops or varieties of tree crops that are self-fertile do not necessarily preclude MP. For example, fruit set of self-fertile almond varieties ('Independence') was shown to be 60% higher in bee-pollinated (20%) than bee-isolated trees (31%) [113]. Similarly, the self-pollinating *Coffee arabica* produces 35% more yield (in weight) when bees can visit flowers [114], lending weight to the potential use of MP systems within these crops.

As shown in Table 4, our understanding of pollination biology, and ecology of the tree crops themselves and their pollinators, is deep for some crops but scant for others (e.g., Cacao [7]). Further, most tree crops are now grown outside their native range due to global demand or benefits borne by comparative advantage. For example, Brazil is currently the world's eighth largest macadamia producer. However, knowledge on the pollination biology of macadamia crops in Brazil was reported to be still developing and insufficient—the main visitor was shown to be butterflies (50% of floral visits) [37] rather than exotic honey bees and native stingless bees, as observed in its native Australian range [8]. Therefore, understanding the pollination biology will be critical in determining the feasibility of MP for specific tree crops. Key factors include pollen type, flower morphology flowering pattern and stigma receptivity, as discussed below.

**Table 4.** Pollination biology of selected fruit and nut crops. Feasibility of mechanical pollination (MP) for each crop is rated as highly feasible (commercial reality, +++), feasible (some evidence available, ++), theoretically feasible (+), or not feasible.

| Tree Crop (Blooming Period) | Pollination Biology | Pollen Type | Natural Fruit Set (%) | Stigma Receptivity Per Flower | Comment | MP Feasible? | References |
|---|---|---|---|---|---|---|---|
| Apple (*Malus domestica*) 14 days | • Self-incompatible for most varieties<br>• Monoecious with perfect flowers<br>• Entomophilous | Orthodox | Only 2–5% required for a commercial crop (crop thinning required) | King flowers—up to 2 days after anthesis; lateral flowers —up to 4 days after anthesis | 4 to 5 seeds require pollination to ensure symmetrical shape. Tendency for biennial bearing of some cultivars. | +++ | [10,115] |
| Apricots (*Prunus armeniaca*) 16 days | • Old European cultivars are mostly self-fertile but those from Central Asia and middle-east and newly bred European cultivars are self-incompatible<br>• Monoecious with perfect flowers<br>• Entomophilous | Orthodox (most cultivars)Recalcitrant (Iranian varieties?) | 41–77% (no pollinator required) | Variable data: 2–4 days after anthesis. Optimal at flat petal stage or exhibiting petal fall | Apricot one of the first fruit tree to flower in early spring | +++ | [71] |
| Avocado (*Persea americana*) 2 days | • Self-fertile, however temporal and spatial separation of male and female promotes cross-pollination<br>• Monoecious with<br>• Entomophilous | Recalcitrant | 0.3% | commonly 3–4 h | | Not feasible (protogynous dichogamy may limit logistics) | [35] |
| Blueberries (*Vaccinium corymbosum*) 14 to 21 days | • Self-incompatible for most varieties<br>• Monoecious with perfect flowers<br>• Entomophilous | Recalcitrant | 50–70% is considered good | 3–5 days after anthesis | Flower morphology prevents pollen from falling onto the stigma | Not feasible (style length is usually shorter than the urn-shaped Corolla, which makes it difficult to access the stigma) | [116] |
| Date Palm (*Phoenix dactylifera*) 14 to 21 days | • Cross-pollination<br>• Dioecious<br>• Anemophilous | Orthodox | 13 to 50% (in commercial practice, all palms are artificially pollination). | Variable: 1–14 days after spathe opening | Unsuccessful pollination results in parthenocarpic fruit, which are inedible | +++ | [117] |

**Table 4.** *Cont.*

| Tree Crop (Blooming Period) | Pollination Biology | Pollen Type | Natural Fruit Set (%) | Stigma Receptivity Per Flower | Comment | MP Feasible? | References |
|---|---|---|---|---|---|---|---|
| Kiwifruit (*Actinidia deliciosa*) 14 to 42 days | • Cross-pollination<br>• Dioecious<br>• Entomophilous | Orthodox | 90% (with managed pollination service) | Receptive for up to 7 days after anthesis but decreases after the fourth day | Flowers are nectarless. 1000 ovules per flower need to be fertilised to produce large fruit | +++ | [55] |
| Mango (*Mangifera indica*) Up to 25 days | • Complete or partial self-incompatible and self-fertile<br>• Andromonoecious<br>• Entomophilous | Recalcitrant | <0.25% | 1 day prior to and 2 days after anthesis but maximum with within 3 h after anthesis | Tendency for biennial bearing | + | [115] |
| Olive (*Olea europaea*) 21 days | • Partially self-incompatible<br>• Andromonoecious<br>• Anemophilous | Orthodox | 1–2% fruit set is sufficient for commercial yield | 4 to 12 days | Tendency for alternate bearing. | +++ | [118] |
| Peach and nectarine (*Prunus persica*) 16 to 25 days | • Mostly self-fertile<br>• Monoecious with perfect flowers<br>• Entomophilous | Orthodox | Up to 58% | Receptive for 3 days | Bees can improve number and size of fruit | +++ | [119] |
| Pear (*Pyrus communis*) Japanese pear (*Pyrus pyrifolia*) 10 to 25 days | • Self-incompatible for most varieties<br>• Parthenocarpy<br>• Monoecious with perfect flowers<br>• Entomophilous | Orthodox | Up to 30%, requires thinning | Receptivity for up to 6 days per flower but within a flower, stigmas may be immature, mature or degenerated. | Low volume of nectar (<3 μL) and its low sugar concentration (<25%). All 5 seeds require pollination to ensure symmetrical shape. | +++ | [120] |
| Plum (*Prunus domestica*) 40 days | • Self-incompatible for most varieties (Japanese and European plums do not cross pollinate each other)<br>• Monoecious with perfect flowers<br>• Entomophilous | Orthodox | Up to 40%, requires thinning | 1 to 4 days after anthesis | Tendency for biennial bearing | +++ | [121] |

**Table 4.** *Cont.*

| Tree Crop (Blooming Period) | Pollination Biology | Pollen Type | Natural Fruit Set (%) | Stigma Receptivity Per Flower | Comment | MP Feasible? | References |
|---|---|---|---|---|---|---|---|
| Sweet cherry (*Prunis avium*) 10 to 24 days | • Self-incompatible for most varieties • Monoecious with perfect flowers • Entomophilous | Orthodox | 15–80%, sometimes requires thinning | Optimal 2–3 days after anthesis | | +++ | [101,122] |
| Almond (*Prunus dulcis*) Up to 1 month | • Self-incompatible for most varieties • Monoecious with perfect flowers • Entomophilous | Orthodox | 30% | Optimal when flowers are past the fully open stage and not at younger stages. | Bloom in late winter and early spring when pollinators are scarce. Tendency for alternate bearing. | +++ | [71] |
| Brazil nut (*Bertholletia excelsa*) Up to 4 months | • Allogamous • Monoecious with perfect flowers • Entomophilous | Recalcitrant? | <1% | 1 day | Requires pollinators that are sufficiently big enough to uncurl the ligule/androecial hood. Largely wild harvested. | Not feasible (flower morphology would prevent delivery of pollen | [123] |
| Cashew (*Anacardium occidentale*) Up to 3 months | • Self and cross-pollination • Andromonoecious • Entomophilous | Orthodox | 10–27% | Up to 1 day after anthesis and peaking between 10 to 12 am. | | ++ | [124] |
| Chestnut (*Castanea sativa*) 1 month | • Most are self-incompatible • Andromonoecious • Duodichogamous • Entomophilous | Orthodox | 24–66% | 1 day pre- and 2 days post-anthesis (each flower has 6–8 stigmas that are receptive one at a time) | Delayed fertilisation of ~ six weeks until ovary is mature | ++ | [125] |
| Hazelnut (*Corylus avellana*) Up to 3 months | • Most are self- incompatible (SSI) • Andromonoecious • Dichogamous • Anemophilous | Recalcitrant | 66–82% | Up to 3 months | Pollination occurs in winter. Delayed fertilisation of up to 3 months until ovary is mature. | ++ | [64,68] |

**Table 4.** *Cont.*

| Tree Crop (Blooming Period) | Pollination Biology | Pollen Type | Natural Fruit Set (%) | Stigma Receptivity Per Flower | Comment | MP Feasible? | References |
|---|---|---|---|---|---|---|---|
| Macadamia (*Macadamia integrifolia*) 33 days in Australia (up to 5 months in Hawaii) | • Self-incompatible • Monoecious with perfect flowers (protandrous) • Entomophilous | Orthodox | 3–4% | 2–3 days after anthesis | Observed xenic effects of kernel size and mass. Bloom in late winter and early spring. | ++ | [8,111] |
| Pecan (*Carya illinoinensis*) Up to 28 days | • Cross-pollination • Andromonoecious • Dichogamous (both protandrous and protogynous cultivars) • Anemophilous | Recalcitrant | 50% | 1 day after anthesis for up to 2 days | | + | [126] |
| Pistachio (*Pistacia vera*) Up to 25 days | • Cross-pollination • Dioecious (often protandrous) • Anemophilous | Recalcitrant | <16.5% | Up to 4 days after anthesis | Tendence for alternate bearing | ++ | [127] |
| Coffee robusta (*Coffea canephora*) Up to 3 months | • Most are self-incompatible • Entomophilous | Orthodox | 9% | | Bees boost yield by up to 40% | + | [128] |
| Cacao (*Theobroma cacao*) Blooms all year | • Most are self-incompatible • Monoecious with perfect flowers • Entomophilous | Recalcitrant | <5% | 2–3 days after anthesis | pollinated by specialised insects <2 to 3 mm i.e., tiny flies | ++ | [9] |

### 5.1. Pollen Type

The fruit and nut crops listed in Table 1 all have binucleate pollen, indicating that they remain viable following drying and, therefore, amenable to long-term storage. However, as reviewed in Pacini and Dolferus (2019) [129], pollen can also be classified on the basis of its water content at dispersal: orthodox and recalcitrant pollen is dispersed in partially desiccated (<20%) or partially hydrated (>20%) forms, respectively. In practical terms, this means that it may be easier to process orthodox pollen, which is more dehydration tolerant than recalcitrant pollen, which is dehydration sensitive. For tree crops with recalcitrant pollen, including avocado, hazelnut, mango, pecan, pistachio and walnut, developing protocols to maintain pollen longevity may be more challenging. Indeed, pollen storage trials confirm that germination of mango, pistachio and pecan was greatly reduced following storage, and it could only be maintained for mango when stored at −196 °C (Table S1). Further, it may be that the observed lack of improvement in fruit set of pistachio following MP was due to poor pollen viability, though no details on pollen quality were provided in the three papers published (Table 2).

### 5.2. Flower Morphology

The success of MP depends on the successful delivery of pollen to the stigma but, in most cases, the delivery method is imprecise. Ideally, the stigma should be completely exposed and unimpeded by any physical barriers to maximise the chances of the pollen landing on its surface. A careful review of the flower morphology indicates this is the case for most tree crops, with a few exceptions. Notably, the flower of Brazil nut is characterised by the presence of an androecial hood that is usually uncurled by the pollinator to access the stigma [123], and the style of blueberry flower is usually much shorter than the urn-shaped corolla, thereby making it difficult for blown pollen to access the stigma (Table 1).

### 5.3. Flowering Pattern and Stigma Receptivity

The critical information required to determine the timing and frequency of MP applications includes temporal flowering patterns at the whole tree level and stigma receptivity at the individual flower level. It can be seen in Table 2 that both bloom duration and stigma receptivity vary greatly amongst tree crops from a few hours to weeks or months. Further variation can occur within a species, cultivar or site, or with environmental conditions [122,130]. Given these variations, stigma receptivity, in particular, is best linked to stage of development rather than a set timeframe (e.g., pecan [126]).

For tree crops with extremely short periods of stigma receptivity, such as avocado cultivars that flower as female in the afternoon and then male in the morning [131], clearly it will be logistically difficult to apply MP at a commercial scale. In contrast, kiwifruit appeared equally able to set fruit at any time of the day, meaning there would be flexibility to apply MP outside of typical pollinator hours [103]. For dichogamous tree crops and, in particular, protandrous cultivars that shed pollen prior to stigma receptivity (Table 1), there is potential to harvest the pollen in the same growing season and therefore utilise fresh pollen that has not been stored. For example, some male cultivars of Pecan shed pollen early in the season and up to two weeks before the stigma is receptive [126]. In its native environment, hazelnut stigmas can remain receptive for a few months; however, hazelnuts grown outside their traditional cultivation areas had greatly reduced receptivity of only 1–2 weeks due to warmer and drier winter conditions [64]. Clearly, the development of MP systems would need to be tailored under local systems and environmental conditions.

### 5.4. Training System

In the past 50 years, fruit orchards have transitioned from the traditional vase training system to higher density trellised two-dimensional training systems. When combined with dwarfing rootstocks, this system can increase early light interception and penetration of light into the canopy [132] and allow for anticipated efficiencies in automation, particularly in harvesting operations and mechanical pruning [133]. More recently, investigations have

been conducted to determine the potential of robotic pollination [134]. Modern orchard systems increase the efficacy of spray applications including MP systems. The narrow and compact canopy of the 2-D system provides a more uniform and easily targeted floral canopy for even dispersal of pollen, thereby reducing the amount of pollen required. In future, the design and management of a canopy structure that maximises pollen delivery may be guided by functional structural plant modelling (FSPM). Advances in FSPM have enabled the development of comprehensive crop models that consider complex factors such as architecture, phenology and physiological processes at the tree scale [135]. FSPM has been successfully used to investigate tree development and fruit production of tree crops such as mango [136] and macadamia [137]. The mango model was based on simulating the appearance of growth units and inflorescences, whereas the macadamia model was based on simulating carbon allocations.

*5.5. Tree Height*

Some fruit and nut trees, particularly where dwarfing rootstocks are not available, can reach heights of up to 30 to 50 m (e.g., macadamia and Brazil nut [138]). Clearly, ground-based pollen delivery is a challenge for these crops. Aerial delivery of pollen using drones is a realised commercial service offered by a US company for crops such as apple, almond, sweet cherry and pear, with a system for dates (23 m tall) currently being developed (e.g., Dropcopter [97]). In drone pollination, the carrying capacity is limited (e.g., 5 kg); therefore, pollen is largely delivered without a carrier (Table 3). Whether drone pollination would be a viable option for such tall tree crops remains unclear; however, it may be limited to wind-free days, which may impact the commercial viability of the MP systems in these crops.

## 6. Conclusions

Pinillos and Cuevas (2008) [34] asked whether MP can replace natural pollination so that only the "best pollen can be selected and applied in the most suitable moment and in an adequate amount to obtain a satisfactory yield." Since then, although new MP systems are a commercial reality for some fruit and nut crops, MP has generally supplemented rather than replaced natural pollination. It is clear that there are a number of challenges impeding the progress and wider uptake of MP systems.

Firstly, our review identified critical gaps in the current knowledge of the pollination biology of economically important fruit and nut crops. These knowledge gaps currently limit the development of MP systems, particularly for emerging nut and fruit species that are grown outside their native distributions. Adding to this are the unknown effects of anthropogenic climate change on plant development and flower phenology (e.g., sweet cherry [139]) and geographic distribution of pollinators (e.g., for coffee [140]; for Brazil nut [138]).

Secondly, results from peer-reviewed papers are, at best, obtained at the tree scale (possibly a reflection of the practical reality of research). Arguably, convincing the horticultural industry about the benefits of MP systems will require up-scaling to demonstrate the applicability of cost and logistics against improved pollination outcomes over space and time.

Thirdly, although outside the scope of this review, there is a need to consider the contribution of wild pollinator species [141] and the increasing use of species that are non-*Apis* species e.g., bumble bee [142] and solitary Osmia bee [6] in almond pollination. Questions remain about how they may affect the efficacy of MP systems, which have been largely developed to supplement the use of European honey bees.

Lastly, given that companies are already selling high-quality pollen and equipment specific to MP systems, and offering MP services, there may be a large disparity between what is published in peer-reviewed papers and the commercial knowledge due to 'in-house' R&D. Lack of access to this proprietorial knowledge is a barrier in extending MP systems to other crops. MP systems have been developed to improve pollination of highly intensive

industries that are largely risk averse. Once reassured that the technology and systems work consistently and are economically viable, then it is envisioned rapid investment and uptake of MP would occur in order to mitigate the risk of crop yield and quality limitations due to lack of effective pollination.

**Supplementary Materials:** The following supporting information can be downloaded at: https://www.mdpi.com/article/10.3390/agronomy12051113/s1. Table S1: Selected studies that examined effect of pollen storage temperature on pollen viability of fruit and nut crops (published since 2008).

**Author Contributions:** Writing—original draft preparation, A.E.; writing—review and editing, D.C.C., S.R.Q., G.R.A., C.J.S., K.M.B., M.D.W. and A.J.G.; funding acquisition, A.J.G. All authors have read and agreed to the published version of the manuscript.

**Funding:** This project is being delivered by Hort Innovation, from the Australian Government Department of Agriculture, Water and the Environment as part of its Rural R&D for Profit program (ST19000) and University of Tasmania, University of Adelaide, Plant and Food Research, University of New England and NSW Department of Primary Industries: "Novel technologies and practices for the optimisation of pollination within protected cropping environments".

**Data Availability Statement:** Not applicable.

**Acknowledgments:** We thank Michele Buntain for valuable comments on an earlier draft of the manuscript.

**Conflicts of Interest:** The authors declare no conflict of interest.

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
