# Peer review of "Feasibility of Mechanical Pollination in Tree Fruit and Nut Crops: A Review"

_agronomy, doi:10.3390/agronomy12051113_

Round 1

Reviewer 1 Report

The paper needs a revision of the cited references. See the attached file for details. The table are not easy to read and need a redrawing (see the attached file for details).

Reviewer 2 Report

At first, I felt somewhat misled by the abstract, mostly because I had no idea why the "intrigue" of business secrets mattered to the actual science. This aside, I was given a lot of good information and a very interesting read.

Erros all concern bees and pollinators, and the authors should 'move up' from the Discussion into the Introduction that this is not a paper that much deal with them, or other pollinators. However, any 'hybrid' system imaginable will have to consider what they actually do, and what they might be doing in the future and, of course, who they actually are and where they live. All these little things matter greatly, and were discussed and given many examples in two volumes of a recent FAO, on-line, free 500+ page publication (2018, The pollination of cultivated plants: a compendium for practitioners).

The few errors in the paper I found were these:

line 81;   the same pollinating bee, Eulaema, mostly E, cingulata, is found from Brazil to Mexico. It is not solely Mexican.

All table 2.  What is fruit set? initiation or full and mature? There are differences in the literature and the biology is very important

also, 'on' and 'off' years may not be familiar to all. It is about parental investment, and also about plant longevity. AND PLANT LONGEVITY IS MOSTLY IGNORED. Both cacao and peach trees usually die after a fruit set approaching 100%

also, almond listed as MP 17% and then 57% for both bee and MP? You mean combined, so in theory 40% improvement via bees? Unclear

line 251. 'is' more differentiated

line 276. grey literature. Can be bogus, not peer reviewed, and no idea why it should matter here. Please clarify if possible.

line 308. why not fungal spores too?, and mention amplicon sequencing far more sensitive than artificial inoculations, and the observations reported are not field tests. Disease transmission at floral 'hubs' is currently ruining a lot of pollination schemes and general 'bee health'

line 322 Vannette?

line 368. Coffee arabica is the best example of a tree that is self-compatible but produces 35% more yield (in weight) when bees can visit flowers (Roubik, Nature 2002).

line 378. Apis is not native there

Table 3.  avocado flowers are attractive to Apis and visited by them, as Geotrigona and Trigona and Scaptotrigona, in Neotropical area, ALSO, there is no 'Meliponinae', it is subfamily Apinae, tribe Meliponini (taxonomy of last 32 years)

454-463  it seem very far-fetched to expect drones to disperse pollen to 40 m trees- why not eliminate this rather sketchy section? Big pollen wastage for any but totally open flowers, totally impractical in most weather

line 465. asked 'whether' MP could
